# Application of the Child Health and Nutrition Research Initiative (CHNRI) methodology to prioritize research to enable the implementation of Ending Cholera: A global roadmap to 2030

**Melissa Ko**[1☯]*, **Thomas Cherian**[1☯], **Helen T. Groves**[2‡], **Elizabeth J. Klemm**[2‡], **Shamim Qazi**[3☯]

1 MMGH Consulting GmbH, Zurich, Switzerland, 2 Wellcome Trust, London, United Kingdom,
3 Independent consultant (Retired World Health Organization staff), Geneva, Switzerland

☯ These authors contributed equally to this work.
‡ These authors contributed to shaping the concept and study design but did not participate in data collection or analysis
* kom@mmglobalhealth.org

**Data Availability Statement:** All results from the final survey are available from the Dryad database

## Abstract

### Background

The "Ending Cholera: A Global Roadmap to 2030" (Roadmap) was launched in October 2017. Following its launch, it became clear that additional evidence is needed to assist countries in controlling cholera and that a prioritized list of research questions is required to focus the limited resources to address the issues most relevant to the implementation of the Roadmap.

### Methods

A comprehensive list of research questions was developed based on inputs from the Working Groups of the Global Taskforce for Cholera Control and other experts. The Child Health and Nutrition Research Initiative methodology was adapted to identify the relevant assessment criteria and assign weights to each criterion. The assessment criteria were applied to each research question by cholera experts to derive a score based on which they were prioritized.

### Findings

The consultation process involved 177 experts and stakeholders representing different constituencies and geographies with research priority scores ranging from 88·8 to 65·7% and resulted in the prioritization of the top 20 research questions across all Roadmap pillars, the top five research questions for each Roadmap pillar, and three discovery research questions. This resulted in 32 non-duplicative research questions that considers both immediate and long-term Roadmap goals.

(doi:10.5061/dryad.h44j0zpkr). All other data is within the Supporting Information files.

**Funding:** MMGH Consulting GmbH (MMGH) was commissioned by the Wellcome Trust on behalf of the Global Task Force on Cholera Control. MMGH responded to a request for proposals and won the bid. Wellcome Trust played a role in shaping the concept and study design but did not participate in data collection, analysis or interpretation or the decision to submit the paper for publication.

**Competing interests:** The authors have declared that no competing interests exist.

## Interpretation

The transparent, inclusive, and rigorous process to develop a Research Agenda is aimed to secure broad buy-in and serve as a guide for funding agencies and researchers to focus their efforts to fill the evidence gaps plaguing cholera-endemic countries.

## Introduction

Cholera is a diarrheal disease that can be treated with rehydration; however, without treatment, cholera can kill within hours. Although cholera has been eliminated from high-income countries for over 150 years, it remains an important public health problem and is endemic mainly in the low- and middle-income countries in sub-Saharan Africa and Asia. Cholera remains a stark marker of inequity and continues to disproportionately affect the poorest and most vulnerable populations around the world and within each affected country. It is currently estimated that cholera affects at least 47 countries and causes 2.9 million cases and 95,000 deaths per year worldwide with large, devastating outbreaks still occurring at regular intervals such as in Zimbabwe 2008, Haiti in 2010, Sierra Leone in 2012, and more recently in Yemen, 2016 [1,2].

In October 2017, the partners of the Global Task Force on Cholera Control (GTFCC) endorsed a call to action to end cholera through the implementation of a new strategy known as "Ending Cholera: A Global Roadmap to 2030" (the Cholera Roadmap). The Cholera Roadmap champions a multi-sector approach focused on cholera hotspots where the populations most affected live and built on the different pillars for cholera control, namely disease surveillance (epidemiology and laboratory confirmation); vaccination with oral cholera vaccines (OCV); water, sanitation, and hygiene (WASH); case management (CM); and community engagement (CE). The Cholera Roadmap aims to achieve a 90% reduction in cholera deaths and cholera elimination in 20 countries by 2030 [2,3].

With the launch of the Cholera Roadmap, it became clear that additional evidence is required on the effectiveness of existing tools and interventions and how to optimise their implementation. However, given the limited resources and competing priorities, it is important to prioritize research to fill the key evidence gaps for optimal implementation of the Cholera Roadmap. A 12-month process was launched to develop a prioritized Research Agenda for the Cholera Roadmap by adapting the Child Health and Nutrition Research Initiative's (CHNRI) approach, which is reported here [4–9]. The development process was guided by a Steering Committee convened specifically for this purpose.

## Methodology

The research prioritization process was conducted in two phases. The first phase consisted of adapting the CHNRI method to achieve alignment with the multi-sector approach of Cholera Roadmap and the second phase utilized the adapted methodology to develop a prioritized list of research questions for inclusion in the Research Agenda.

### Phase 1

Phase 1 focused on identifying the relevant research questions for inclusion in the prioritization exercise and adapting the CHNRI methodology by defining the Research Agenda's context and selecting and weighting the most appropriate prioritization criteria.

The potential research questions were identified through three main sources (i) the GTFCC's five working groups (ii) stakeholder interviews and survey and (iii) publicly available documents such as the World Health Organization (WHO) research priorities for cholera vaccination. These questions were reviewed, consolidated and classified according to the relevant Roadmap pillar and the CHNRI 4D framework categories i.e., Delivery, Development, Description and Discovery. In some situations, if research questions were relevant to more than one Roadmap pillar, they were classified as "cross-cutting" [5]. The consolidated questions underwent an iterative consultative process with 20 cholera experts to remove duplication and redundancies, standardize the language and format to improve clarity and facilitate scoring of the questions to arrive at a final set of research questions for prioritization. Four hundred and fifty-three research questions were identified by the GTFCC working groups and through the interviews and surveys. These were reviewed by GTFCC working group chairs and key technical experts and consolidated into 124 questions after removal of duplications and non-research statements. These 124 questions underwent further review by 17 cholera experts, and ultimately, 93 were chosen for inclusion in the prioritization process. Thirty-one questions were excluded based on feedback from the experts that sufficient evidence and guidance already existed for those questions. The 31 research questions were flagged to the GTFCC for follow up to determine if systematic reviews and grading of the quality of the evidence was needed to determine the need for additional research to generate primary data. The list of 93 research questions selected for prioritization included 25 (27%), 23 (25%), 19 (20%), 16 (17%), and ten (11%) classified as cross-cutting, epidemiology, surveillance and laboratory, case management, OCV, and WASH pillars, respectively. Given their cross-cutting nature and the importance in all cholera interventions, the ten community engagement questions were all classified into the cross-cutting category. Per the 4D category, the distribution of the 93 questions were split amongst the 4D domains of delivery, development, description, and discovery was 36 (39%), 35 (38%), 19 (20%), and 3 (3%), respectively.

In parallel, stakeholder consultations were held (via online survey and telephone interviews) to select the most important research prioritization criteria. Eight potential criteria were selected for consultation including affordability, ethical answerability, equity, fundability, impact, implementability, relevancy, and sustainability. Inputs were obtained to agree on the definition of each criterion and its weight in the prioritization process as well as on the importance of each criterion independent of the others. Four hundred and nineteen individuals were contacted to either complete an online survey via Qualtrics™ (n = 306) or participate in a 45-minute telephone interview (n = 113). The telephone interviews specifically targeted key senior-level experts and country-level stakeholders who were considered less likely to respond to an online survey and could provide more qualitative feedback that would be difficult to capture in a survey.

Both the interviewees and survey respondents were asked four demographic questions and to rate the eight proposed criteria and their description using a 5-point Likert scale of "not important at all" to "extremely important". A mean was calculated for each criterion based by applying points of one for "not important at all" to five for "extremely important", which was used to narrow the number of prioritization criteria. This rating was used to ultimately select 5 prioritization criteria for us in the final prioritization exercise. Further, the respondents were given open-ended questions to describe their rationale for their rating, improve the criteria descriptions, and indicate what they considered were key evidence gaps. All qualitative responses were combined and analyzed through an iterative process with the interviewer to develop key themes. Additional stratified analyses were conducted to evaluate any trends in the responses based on the demographic characteristics. See S1 File for the interview and survey questions.

One hundred and forty-one experts representing 32 countries provided their feedback either through the interviews or the online survey in Phase 1, informing in the definition of the Research Agenda context and selection of five criteria for use in prioritizing the research questions. The definitions of the five selected criteria were revised based on the feedback from the interviews and survey to improve clarity and enable uniform use in the prioritization process. Following the CHNRI guideline, the Research Agenda context was defined as follows **(i) Population of interest**: all countries and communities where cholera is endemic and/or there is an epidemic risk of cholera **(ii) Time scale**: present-day to 2030 **(iii) Geographic scope of research**: global, regional, national, and sub-national levels. Sub-national may include different administrative levels, such as provinces or states, districts, communities or households **(iv) Impact of interest**: reduction of deaths and burden of cholera where burden may include prevalence and morbidity as well as any economic or social impact of cholera. Additional information may be found in S2 File.

Following the initial analysis which identified 5 prioritization criteria and to finalize the CHNRI approach, a virtual meeting was conducted with a subset of the stakeholders who participated in the interviews and survey to discuss the design of the survey, Research Agenda's context, and weighting of each criterion. To inform this discussion and determine appropriate weights to each of the five criteria finally selected for the prioritization exercise, a second online survey was sent to 85 stakeholders to "distribute 100 points across the five criteria according to the perceived level of importance, i.e., to allocate higher points to the criterion considered as the most important". This process enabled the weighting of each criterion based on its importance relative to that of the other criteria. Following the CHNRI approach, the weights were calculated by dividing the mean values allocated to each criterion by 20 or the value if each of the 100 points was distributed equally between the 5 criteria [10]. Of the 85 stakeholders, 40 responded to the second survey. This method allowed the respondents to rank each criterion in respect to other criteria, which was used to set the appropriate weights for each criterion. Table 1 provides an overview of the five criteria and their weights to evaluate the research questions.

## Phase 2

An online survey was designed using Qualtrics™ to allow respondents to score the identified research questions using the chosen criteria. The survey asked three demographic questions related to respondent location, organization affiliation, and areas of expertise. The research questions were organized into "blocks" by the Roadmap pillar, including case management, community engagement, OCV, WASH, and cross-cutting (if the research questions were

**Table 1. Five criteria utilized to evaluate research questions.**

| Criterion | Weight | Description |
|---|---|---|
| Answerability | 0·79 | Do you think the proposed research is answerable in cholera-affected countries and communities? *Assumes all protocols will be subject to appropriate ethics reviews. |
| Impact | 1·20 | Will the research outputs contribute to reducing cholera deaths and burden? *Burden may include morbidity, economic or social impact |
| Implementability | 1·12 | Will the proposed research lead to solutions that are implementable (e.g. feasibility of introduction, including acceptability to the cholera-affected communities and scale-up)? |
| Relevancy | 1·06 | Will the proposed research contribute to addressing relevant evidence gaps in the cholera-affected countries or communities when implementing the Cholera Roadmap? |
| Sustainability | 0·83 | Will the proposed research lead to solutions that are sustainable over time without, or with only limited, external financial or technical support in cholera-affected countries? |

relevant to more than one pillar). The blocks were randomized for each survey participant to allow for the inclusion of partial responses without bias towards one Roadmap pillar. The respondents were given options of "Yes", "No", and "Maybe" for questions that they felt they could answer, which were assigned points of one, zero, 0·5, respectively. They also had the option to answer "don't know" for those question that they felt were outside their area of expertise and knowledge. "Don't know" responses were excluded from the analysis to calculate research priority scores. See S3 File.

Two hundred and forty-five individuals were sent personalized emails to complete the survey. Further, a link to an anonymous survey was also posted on the GTFCC's website. The online survey remained open for two months with regular email reminders sent weekly.

All responses were downloaded into Microsoft Excel from the Qualtrics™ software, including any partially completed surveys where a response was provided to at least one research question. The following scores were calculated for each research question:

- **Unweighted research priority score (RPS)**: the following formula was used where $c$ is the five criteria evaluating the research question.

$$\text{Unweighted RPS} = \frac{1}{5} \times \sum_{c=1}^{5} \frac{(N_{Yes} \times 1) + (N_{Maybe} \times 0.5)}{N_{Yes} + N_{No} + N_{Maybe}}$$

- **Weighted research priority score:** the following formula was used, where $W$ is the weight for each criterion and $c$ is the five criteria evaluating the research question. The following weights were applied 0·79, 1·20, 1·12, 1·06, and 0·83 for Answerability, Impact, Implementability, Relevancy, and Sustainability, respectively.

$$\text{Weighted RPS} = \frac{1}{5} \times \sum_{c=1}^{5} W_c \times \frac{(N_{Yes} \times 1) + (N_{Maybe} \times 0.5)}{N_{Yes} + N_{No} + N_{Maybe}}$$

Finally, the average expert agreement (AEA) was calculated as well as stratified analyses were also performed using Microsoft Excel to identify any biases considering the respondents' identified areas of expertise and respondent location [11]. In addition to the CHNRI approach defined above, the trends of the responses provided were also analysed manually, particularly the "Don't Know".

## Results

### Phase 2

**Identified research priorities.**   One hundred and thirty-eight individuals representing 39 countries scored the 93 research questions as part of Phase 2. Of these, 21 individuals only provided partial responses to some but not all the research questions. These were included in the analysis. Table 2 provides an overview of the demographics of individuals who responded.

Based on the weighted research priority scores, the top 20 research priorities consisted of nine (45%), five (25%), three (15%), two (10%), and one (5%) for OCV, cross-cutting, WASH, epidemiology /surveillance/ laboratory, and case management (CM) pillars, respectively (Table 3). Fourteen questions (70%) were related to Delivery and six (30%) to Development, whereas no Description or Discovery questions ranked within the top 20. From consultation with the Steering Committee and to ensure linkages to the GTFCC and its working groups and their mandates, the top five research priorities of each Roadmap pillar were also identified as key priorities, which resulted in nine additional priorities not captured in the top 20.

**Table 2. Demographics of individuals who evaluated the research questions.**

| | # | % |
|---|---|---|
| **I. Expertise** Respondents were allowed to select up to two areas of expertise | | |
| Epidemiology / Surveillance / Laboratory | 75 | 34% |
| Oral Cholera Vaccine | 55 | 25% |
| Water, Sanitation, and Hygiene | 43 | 19% |
| Case Management | 22 | 10% |
| Community Engagement | 21 | 9% |
| Other | 7 | 3% |
| **II. Organisation Type** | | |
| Impl. partner (US Center for Disease Control, International Organizations, United Nations, Civil Society Organisation, and Non-Governmental Organistaion) | 58 | 42% |
| Academic / Research | 46 | 33% |
| Donor | 15 | 11% |
| Government in cholera endemic countries | 16 | 12% |
| Independent | 3 | 2% |
| **III. Respondent location** | | |
| Global, includes World Health Organization European and American Regional Offices, excluding Haiti | 72 | 52% |
| World Health Organization African Regional Office | 33 | 24% |
| World Health Organization Southeast Asia Regional Office | 20 | 14% |
| World Health Organization Eastern Mediterranean Regional Office | 7 | 5% |
| Haiti | 2 | 1% |
| World Health Organization Western Pacific Regional Offices | 4 | 3% |
| Total | 138 | |

Furthermore, given the importance of Discovery related research questions to long-term elimination goals, the three questions were also included among key priorities. The final selection of the 32 priorities (top 20 key priorities plus non-duplicative top five Pillar priorities) resulted in the selection of ten (31%) in the OCV pillar, seven (22%) in the epidemiology, surveillance and laboratory, and five each (16% each) in WASH pillar, cross-cutting, and case management pillars, respectively. Considering the 4D framework, 16 Delivery (50%) and 11 Development (34%) questions accounted for the majority of the priorities selected. The results of the "don't know" analyses did not reveal any significant effects on the overall analyses.

S1 Table provides the full list and scores for the 93 research questions, including the AEA scores.

**Results stratified by the expertise and geographical location.** Stratified analyses were conducted considering the respondents' areas of expertise and location. Several differences were observed in the prioritization of the research questions in the stratified analyses. Differences were observed in prioritization based on the area of expertise of the scorer. For example, the CM experts ranked more CM questions (n = 13) as high importance and WASH experts placing a higher priority on cross-cutting questions (n = 16) (Fig 1). In addition, when considering the respondent's geographical location, individuals based in cholera-endemic countries placed a higher priority on cross-cutting questions (n = 16) and less priority on OCV research questions (n = eight) compared to those at the global level (n = nine and n = 12, for cross-cutting and OCV research questions, respectively) (Fig 2). There were also differences in responses based on geographic regions with respondents in Africa placing a higher priority on cross-cutting questions compared to Asia, which placed a higher emphasis on CM questions (Fig 3). In comparison, there were no significant differences in the 4D categorization of

**Table 3. Priorities for the Cholera Roadmap Research Agenda (n = 32).**

| 4D | Pillar | RQ | Weighted RPS | AEA |
|---|---|---|---|---|
| Delivery | OCV | What are the optimal oral cholera vaccine schedules (number of doses and dosing intervals) to enhance immune response and clinical effectiveness in children 1 to 5 years of age? | 88.8% | 80.7% |
| Delivery | OCV | What are potential delivery strategies to optimise oral cholera vaccine coverage in hard-to-reach populations (including during humanitarian emergencies and areas of insecurity)? | 87.4% | 75.9% |
| Delivery | OCV; WASH | Is there additional benefit to adding WASH packages, for example household WASH kits, to an oral cholera vaccine campaign? | 87.1% | 77.2% |
| Delivery | OCV | What is the optimal number of doses of oral cholera vaccine to be used for follow up campaigns in communities previously vaccinated with a 2-dose schedule? | 86.9% | 76.4% |
| Delivery | OCV | Can the impact of oral cholera vaccine on disease transmission, morbidity and mortality be maximized by targeting specific populations and/or targeted delivery strategies? | 86.8% | 78.0% |
| Delivery | CM | What are the barriers and enablers for integrating cholera treatment into community case management by community health workers? | 86.8% | 74.7% |
| Delivery | WASH | What levels of coverage for relevant water, sanitation, and hygiene interventions is required in cholera hotspots to control and ultimately eliminate the risk of cholera? | 86.3% | 74.9% |
| Delivery | OCV | What impact does the timing of oral cholera vaccine use have on outbreak prevention and control? | 86.2% | 73.9% |
| Development | Epi / Sur / Lab | What is the impact of early diagnosis of cholera using a rapid diagnostic test at the point of care in a community setting compared to testing only in health facilities? | 86.1% | 74.6% |
| Delivery | OCV | How can the use of oral cholera vaccine in the controlled temperature chain (i.e. outside the cold chain) be leveraged to maximize the coverage or impact of vaccination in a field setting? | 85.9% | 75.2% |
| Delivery | All | What is the incremental benefit of implementing a comprehensive interventions package (including water, sanitation, and hygiene, antibiotics, oral cholera vaccine, oral rehydration therapy) to reduce cholera mortality during an epidemic? | 85.7% | 74.3% |
| Delivery | OCV | What is the effectiveness and impact of different vaccination strategies for rapid response to cholera outbreaks (e.g., ring vaccination, case-area targeted interventions, etc)? | 85.3% | 74.1% |
| Development | OCV; WASH | What is the most cost-effective package of water, sanitation, and hygiene and oral cholera vaccine in different situations, based on transmission dynamics in cholera hotspots? | 85.2% | 73.7% |
| Development | WASH | What are the most essential (or what is the minimum set of) infection, prevention, and control (IPC) interventions in cholera treatment facilities and oral rehydration points to reduce risk of transmission within these facilities? | 84.9% | 74.2% |
| Delivery | OCV | Are there immunisation strategies other than repeated mass campaigns that will be effective in preventing endemic or epidemic cholera? | 84.9% | 71.4% |
| Delivery | All | What is the role and added value of CORTs (community outreach response teams) in enhancing case investigation and outbreak detection? | 84.6% | 71.2% |
| Development | OCV | Can oral cholera vaccine be co-administered safely and without interference with other vaccines during mass campaigns or during routine immunization visits (measles containing vaccines, yellow fever, typhoid, meningitis, pneumococcal conjugate vaccine)? | 84.3% | 72.0% |
| Delivery | WASH; CE | What are effective strategies to scale up the use of household water treatment in controlling cholera outbreaks? | 84.1% | 70.9% |
| Development | Epi / Sur / Lab | How can we improve and fine-tune hotspot definition and identification at a district and sub-district level, such as micro-hotspots, including incorporating a population-based approach? | 84.1% | 72.1% |
| Development | WASH | Is improved access to safe water (e.g., water points and distribution networks) effective in controlling and preventing cholera outbreaks? | 84.0% | 74.1% |
| Development | CM | What effect does treatment with antibiotics have on cholera transmission? | 83.3% | 71.0% |
| Development | CM | What is the optimal treatment schedule for antibiotic prophylaxis given to household contacts of cholera patients and does this have an effect on the magnitude, transmission and secondary attack rate of cholera outbreaks? | 80.5% | 69.7% |
| Description | CM | What are the common cholera treatment complications in vulnerable populations (for example: pregnant women, the elderly, those with severe acute malnutrition)? | 80.2% | 66.5% |
| Development | CM | Would ReSoMal formulated with higher sodium, or standard oral rehydration solution containing high potassium, result in lower mortality or morbidity, compared to the standard WHO rehydration solution, in children with severe acute malnutrition? | 80.1% | 65.7% |
| Development | Epi / Sur / Lab | What are the optimal design(s) of surveillance systems (e.g., indicator-based, event-based, community-based, environmental, sentinel site surveillance) to monitor progress of the Cholera Roadmap? | 83.8% | 71.0% |
| Development | Epi / Sur / Lab | What are the optimal surveillance tools (e.g., laboratory methods, case definitions, etc.) to monitor progress of the Cholera Roadmap? | 82.8% | 69.6% |

*(Continued)*

**Table 3.** (Continued)

| 4D | Pillar | RQ | Weighted RPS | AEA |
|---|---|---|---|---|
| Description | Epi / Sur / Lab | How can combined epidemiological and genomic analysis of *V. cholerae* be used to better understand transmission dynamics and inform epidemiological models? | 81.2% | 67.6% |
| Delivery | WASH | How can "design thinking" be used to improve the delivery / uptake of water, sanitation, and hygiene interventions? Design thinking focuses on understanding the needs of the people who will use the intervention and working with them to improve it. | 83.0% | 67.4% |
| Delivery | WASH | What are the factors / determinants that lead to sustainable investments in water, sanitation, and hygiene at country level? | 80.3% | 65.9% |
| Discovery | Epi / Sur / Lab | Research and development of novel and innovative diagnostic tests to accelerate the achievement of the Cholera Roadmap goals. | 80.8% | 65.9% |
| Discovery | OCV | Research and development of new or improved vaccines to contribute to accelerate the achievement of the Cholera Roadmap goals. | 79.5% | 64.7% |
| Discovery | Epi / Sur / Lab | Research to contribute to the collection of genomic data to create a global *V. cholerae* sequences database to map long-range transmission routes. | 72.8% | 55.7% |

prioritized questions with Delivery and Development questions continuing to be prioritized over Discovery and Description.

## Discussion

The 32 questions provide a fairly balanced set of priorities across all of the Roadmap pillars that considers immediate and long-term goals. There was a higher selection of OCV-related questions, which may be related to potential sampling e.g., a high number of stakeholders were involved in OCV-related activities or interpretation bias e.g., OCV tends to provide more concrete impact compared to the other Roadmap pillars. Further, the identified priorities lean heavily towards Delivery and Development research questions as opposed to Description and Discovery, implying a high perceived need to address immediate barriers to implementing the

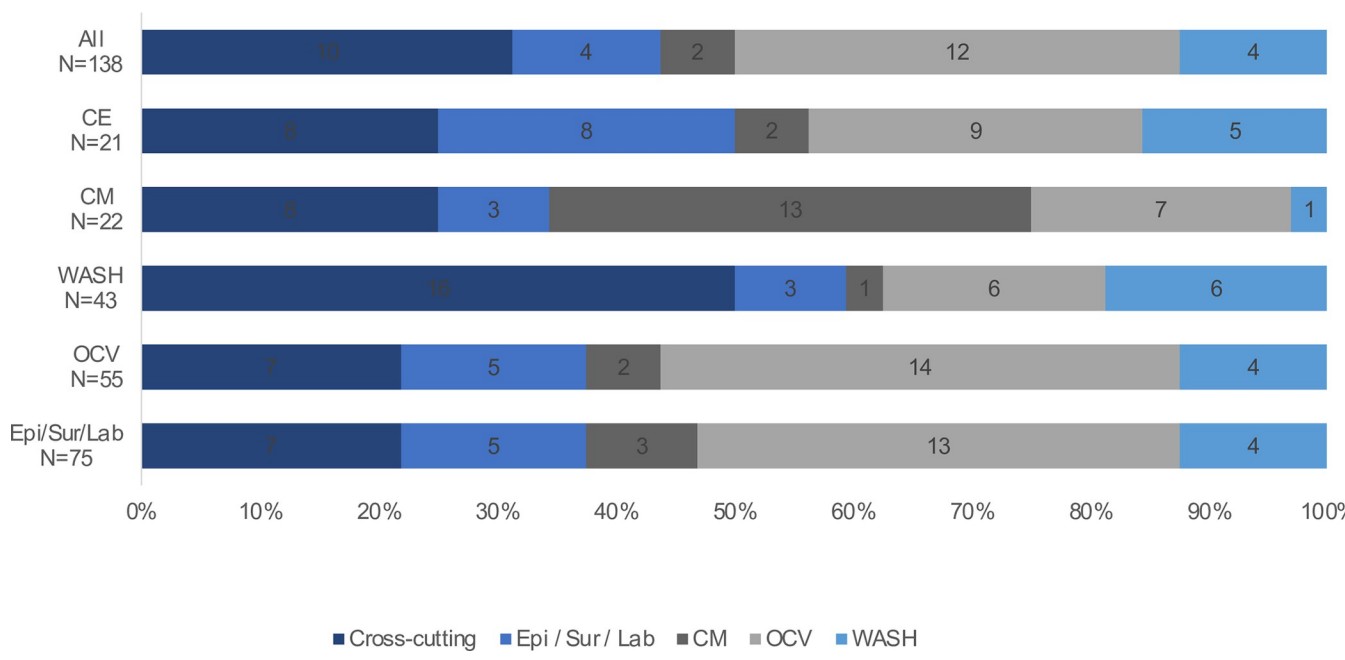

**Fig 1. Breakout of the top 32 priorities by Roadmap pillar and respondent expertise.**

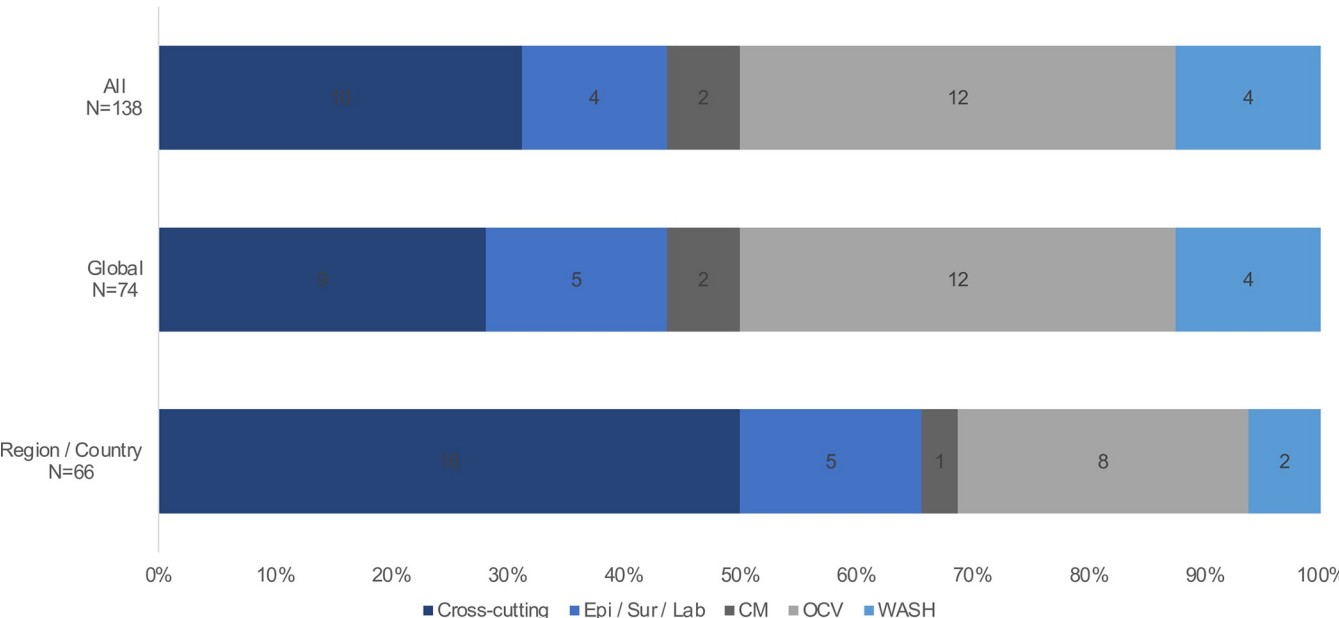

**Fig 2. Breakout of the top 32 priorities by perspective.**

Roadmap interventions. Although the Discovery questions scored lower, they were ultimately included in the final list of priorities based on expert judgement. Even though the Discovery questions may not have outputs prior to the 2030 Roadmap goals, it was felt that the immediate availability of funding could accelerate the availability of innovative solutions that are likely to have an impact on the longer-term goals for cholera control, especially achieving and sustaining elimination.

The stratified analyses clearly demonstrated differences in opinion between the different stakeholder groups, based on their areas of expertise and geographical location. The

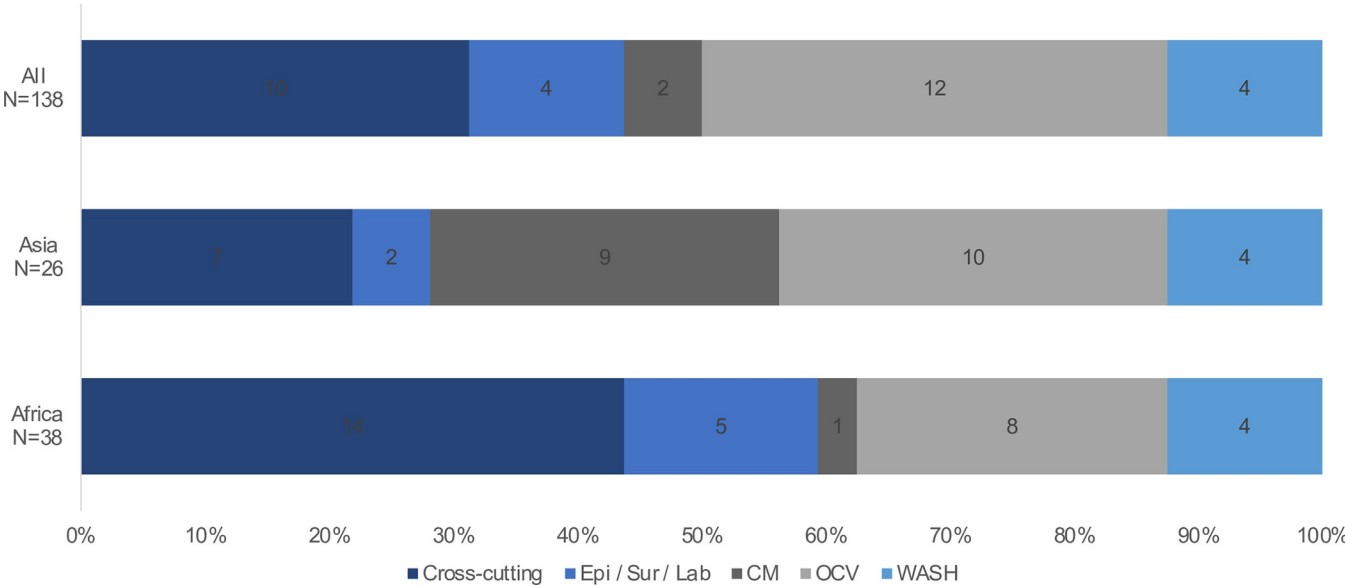

**Fig 3. Breakout of the top 32 priorities by geographic region.**

respondents tended to give higher scores to research questions in their own areas of expertise compared to other areas, which may be driven by their knowledge of the specific Roadmap pillar. If this exercise were repeated, additional efforts should be made to ensure that each Roadmap pillar was equally represented among those participating in the prioritization exercise. When considering the respondent's geographical location, there was a clear divergence in perceptions of the key priorities from those based at the global level compared to those based in cholera-endemic countries. Those located in cholera-endemic countries prioritized more cross-cutting questions compared to those based at the global level. In addition, differences were observed between respondent in different geographical locations. For example, the respondents located in Africa placed a higher priority on cross-cutting questions compared to Asia, which placed a higher emphasis on CM questions. This may reflect the different needs based on the country's current progress towards cholera control and prevention efforts. Further, this reflects the importance of ensuring open dialogue between all relevant stakeholders including the governments where the research priorities will be evaluated to ensure the utilization of research outputs and alignment with ongoing government programmes. Finally, regardless of the stratified analysis, there was general agreement amongst the respondents in all stakeholder groups that the top ten key research questions were very important but less agreement on those ranked questions ranked as 11–32. This indicates that there is more convergence in opinion for the highest priority questions per the AEA, while the lower scores were driven more by a difference in opinion between the different stakeholder groups, rather than a convergence in opinion that these do not represent important evidence gaps. While the stratified analyses provide interesting perspectives on the goals of various stakeholders, the number of individuals for some of the stratified groups, including CE, CM, stakeholders located in Asia and Africa was lower than the threshold of 45 experts required to achieve an optimal collective opinion by CHNRI method [6,12]. Thus there is a risk of bias here because of a small group of respondents whose view of priorities may be influenced by their own knowledge and experience [13,14].

The CHNRI approach was selected amongst other options to identify the cholera research priorities as it is systematic, consultative, transparent, and reproducible [13,15–18]. It incorporates the consideration of values of a wider group of stakeholders. It reduces the impact of self-interest when deriving the initial research question list. Individual ranking reduces any undue individual influence on the process and outcome. In this exercise it allowed the engagement of 177 individuals in identifying research questions, selecting the key criteria and relevant weights to evaluate the research questions, and evaluating the research questions. The process has its limitations also—it is long and sometimes complex, which can affect response rates [19]. If care is not taken to include the government officials in identifying their problems, it may neglect considering the existing government priorities [17]. Finally, it is challenging to obtain the right mix of stakeholders depending upon the area to be explored [14].

In addition to some of the potential biases indicated in the stratified analysis, the other limitation of this work was the suboptimal representation from individuals working for the governments of cholera-endemic countries in identifying their problems and scoring the research questions. Due to COVID pandemic, several face-to-face meetings planned with the government representatives from the cholera-endemic countries were cancelled and their input into the process was affected. Effort was made to involve them through telephone calls, but it was difficult to get time from many of them because they were involved in the response to the COVID pandemic. Second, a systematic literature review was not conducted to identify research gaps. The research questions were largely collected from the GTFCC working groups and supplemented via interviews and consultations, which resulted in a different number of research questions across the Roadmap pillars that were collected via different methodologies

and had different levels of specificity. While efforts were made to standardize the questions, the questions could have been further refined and standardized.

Key strengths of this work included the extensive consultations with stakeholders operating in cholera-endemic countries across different areas of responsibility including policy and decision-makers, donors, and operational leads. This allowed the ability to identify research priorities that are most important for the successful implementation of the Cholera Roadmap.

## Supporting information

**S1 Table. List of 93 research questions and their scores.**
(DOCX)

**S1 File. Interview and survey questionnaire used to adapt CHNRI approach and identify research questions.**
(DOCX)

**S2 File. Phase 1 results.**
(DOCX)

**S3 File. Guidance document accompanying prioritization survey to evaluate research. Questions.**
(DOCX)

## Acknowledgments

The authors thank the 177 experts who actively participated in the exercise by submitting questions, providing feedback, and scoring questions. Special thanks and acknowledgment for the time commitment of Research Agenda Steering Committee, including Dominique Legros, Abul Kalam Azad, Philippe Barboza, Jan Holmgren, Jose Paulo Langa, Daniele Lantagne, and Margot Nauleau, who provided invaluable guidance and feedback on tailoring the process and refining the research questions.

## Author Contributions

**Conceptualization:** Melissa Ko, Thomas Cherian, Helen T. Groves, Elizabeth J. Klemm, Shamim Qazi.

**Data curation:** Melissa Ko, Thomas Cherian.

**Formal analysis:** Melissa Ko, Thomas Cherian, Shamim Qazi.

**Funding acquisition:** Melissa Ko, Thomas Cherian.

**Methodology:** Melissa Ko, Thomas Cherian, Helen T. Groves, Elizabeth J. Klemm, Shamim Qazi.

**Project administration:** Melissa Ko.

**Supervision:** Melissa Ko.

**Validation:** Melissa Ko, Thomas Cherian, Shamim Qazi.

**Visualization:** Melissa Ko, Thomas Cherian.

**Writing – original draft:** Melissa Ko.

**Writing – review & editing:** Thomas Cherian, Shamim Qazi.

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
