## [Decision Letter · Decision Letter 0]

24 Nov 2021

PONE-D-21-25214Application of the Child Health and Nutrition Research Initiative (CHRNI) to prioritize research to enable the implementation of Ending Cholera: A global roadmap to 2030.PLOS ONE

Dear Dr. Ko,

Thank you for submitting your manuscript to PLOS ONE. After careful consideration, we feel that it has merit but does not fully meet PLOS ONE’s publication criteria as it currently stands. Therefore, we invite you to submit a revised version of the manuscript that addresses the points raised during the review process.

We look forward to receiving your revised manuscript.

Kind regards,

Jai K Das

Academic Editor

PLOS ONE

Journal Requirements:

Reviewers' comments:

Reviewer's Responses to Questions

**Comments to the Author**

1. Is the manuscript technically sound, and do the data support the conclusions?

Reviewer #1: Yes

Reviewer #2: Yes

2. Has the statistical analysis been performed appropriately and rigorously? 

Reviewer #1: I Don't Know

Reviewer #2: Yes

3. Have the authors made all data underlying the findings in their manuscript fully available?

Reviewer #1: Yes

Reviewer #2: Yes

4. Is the manuscript presented in an intelligible fashion and written in standard English?

Reviewer #1: Yes

Reviewer #2: Yes

5. Review Comments to the Author

Reviewer #1: This paper presents a research priority setting exercise, using the CHNRI method, to support the Global Roadmap to 2030 for cholera. This paper contains important research priorities and novel adaptations of the CHNRI method. Congratulations to the authors on the manuscript.

I have several questions about the exercise:

- What method did the experts use to further cut down the RQs into a list of 93? Were all the RQs removed answerable through a systematic review or already had sufficient evidence? For those that were answerable through a review, what is the rationale for eliminating systematic reviews (as if called for, they may summarise evidence in a way that is actionable/could lead to policy changes)? Is the exercise intending to look at primary data collection?

- What was the rationale for choosing different methods to weight the criteria in the first and second round? I understand the first round was intended to eliminate criteria through low weights. How did the weights differ with both methods? Was there any lack of engagement doing this exercise twice?

- The discussion states that scorers were more likely to score RQs in their area of expertise higher. Do you have any recommendations for countering this in future (perhaps by either aiming to achieve fairly equal representation across areas of expertise, or only having RQs scored by experts in that area)?

- Were there any partial scores for the list of RQs (e.g., someone started and stopped scoring)? If so, were these included in the RPS? Was there a threshold to the proportion of RQs scored in order to be included?

- What is the rationale for not including Average Expert Agreement?

Some minor comments:

- On line 139-140, should 'research' be capitalised before Agenda (or both lowercase?)

- On line 265, CHNRI acronym has a typo

- It would be useful to spell out all acronyms in their first instance (e.g., OCV, CM, etc.) for readers who are unfamiliar with the acronyms.

Reviewer #2: The authors have tried to create a prioritized list of research questions to focus the limited resources and address the issues most relevant to the implementation of the Roadmap, for funding agencies and researchers to focus their efforts to fill the evidence gaps plaguing cholera-endemic countries. Intense work has been put in through various methods in achieving the research questions, however there are number of limitations and few methods lack clarity.

Most importantly the prevalence of cholera infection and the severity (mortality) due to this infection in low resource countries needs to be mentioned.

While seasonal episode of cholera is most seen, the intervention should focus accordingly. The population that is most affected must be described and strategies to curtail the infection needs to be explicitly explained.

Information on whether the vaccination has helped in reducing the infection rate and the severity must be discussed before implementing to the general population.

As one of the factors mentioned is malnutrition, access to health care facilities and good nutrition, WASH, it is also important to align with the government programs and policies laid for the communicable diseases specific to the country. This is important since the countries selected are a mixed bag with different facilities and opportunities.

There are number of abbreviated terms that needs expansion when used for the first time in the text. Eg: GTFCC’s, OCV

6. PLOS authors have the option to publish the peer review history of their article (what does this mean?). If published, this will include your full peer review and any attached files.

Reviewer #1: **Yes: **Kerri Wazny

Reviewer #2: No

---

## [Author Response · Author response to Decision Letter 0]

7 Jan 2022

7 January 2021

Jai K Das

Academic editor PLOS ONE

Dear Dr. Das, 

We thank you and the reviewers for the informative and helpful assessment of our manuscript. Because of the positive nature of the review, and at your request, we are submitting a revised manuscript. The following is a detailed list of the revisions and/or our responses.

We believe that our responses to the helpful advice from the reviewers and subsequent edits have improved the manuscript substantially. We hope that you will now deem it acceptable for publication. We look forward to your further consideration. Please do not hesitate to contact me with any further questions.

Sincerely,

Melissa Ko, on behalf of the authors

5. Review Comments to the Author

Reviewer #1: This paper presents a research priority setting exercise, using the CHNRI method, to support the Global Roadmap to 2030 for cholera. This paper contains important research priorities and novel adaptations of the CHNRI method. Congratulations to the authors on the manuscript.

I have several questions about the exercise:

- What method did the experts use to further cut down the RQs into a list of 93? Were all the RQs removed answerable through a systematic review or already had sufficient evidence? For those that were answerable through a review, what is the rationale for eliminating systematic reviews (as if called for, they may summarise evidence in a way that is actionable/could lead to policy changes)? Is the exercise intending to look at primary data collection?

The research questions were reduced to 93 based on the feedback of 17 world renowned cholera experts who indicated that the research questions already had a sufficient level of evidence and/or guidance. The 31 research questions that were removed were also flagged for the GTFCC to follow up with to determine if the research question was fully addressed with the available data or whether a systematic review of the available data is required to address the knowledge gap. Please see the revisions in lines 123-129. 

A primary goal of the Research Agenda was to prioritize the research questions that require the generation of primary data to fill existing evidence gaps. As these 31 research questions were deemed to either have sufficient evidence and / or available guidance they were eliminated from the prioritization process but, as we agree with the reviewer that systematic reviews can be very valuable for summarizing evidence in a way that is actionable and can lead to change, we highlighted these research questions as “knowledge” rather than evidence gaps to the GTFCC and its partners for them highlighted as actionable for GTFCC and its partners to follow up on by either communicating the knowledge, developing relevant guidance/technical notes, and determining if additional systematic review should be conducted. 

- What was the rationale for choosing different methods to weight the criteria in the first and second round? I understand the first round was intended to eliminate criteria through low weights. How did the weights differ with both methods? Was there any lack of engagement doing this exercise twice?

The first method focused on rating eight potential criteria (selected on prior CHNRI exercises) that could be used to prioritize the research questions using a Likert scale. The respondents were asked to rate the criteria independently of each other and, as the reviewer notes, this was helpful for eliminating criteria that were deemed less important or relevant for cholera research by the respondents. The first method also allowed respondents to give feedback on the definitions of their preferred prioritization criteria, which we used to refine the definitions for the prioritization exercise to maximize clarity and universal understanding. Following method 1, a set of 5 criteria were selected with some revisions for the prioritization exercise. The second method focused on asking the respondents to rank each of the criteria against the other criteria. The ranking based on relative importance of each criterion allowed for the generation of weights to generate the priority score for each research question. There were slight differences between the two methods. For example” implementability” rated highest, followed by the “impact” in the first method. However, in the second method the ranking of “implementability” and “impact” were reversed. However, the definitions underwent some revisions between the two methods and the definitions for the criteria were not identical between the two rounds. We had about a 50% response rate for both methods. 

Please see lines 140-142, 163-164, 173-176 for further revisions.

-The discussion states that scorers were more likely to score RQs in their area of expertise higher. Do you have any recommendations for countering this in future (perhaps by either aiming to achieve fairly equal representation across areas of expertise, or only having RQs scored by experts in that area)?

Yes, if this exercise were repeated it would be imperative to ensure there is an equal representation of expertise across the different Roadmap pillars. Due to the multi-sectoral approach of the Roadmap and since some individuals may have practical and extremely relevant experiences across multiple pillars but may not identify themselves as an expert in each pillar, we did not want to limit the respondents’ opportunities to prioritize any research questions. 

Line 335-337 have been revised to reflect this information.

- Were there any partial scores for the list of RQs (e.g., someone started and stopped scoring)? If so, were these included in the RPS? Was there a threshold to the proportion of RQs scored in order to be included?

Yes, partial scores were included if they responded to at least one assessment of a research question, please refer to lines 224-226 and 235-237 for the methodology. Note this information was already included so there are no revisions.

There were 21 partial responses recorded and included in the RPS, this has been revised as part of lines 258-259.

- What is the rationale for not including Average Expert Agreement?

The average expert agreement was originally included as part of the supplementary table 4, we did not include it in the main body of the article because there was little discrimination between the AEAs for the research question. However, based on this comment we have edited to include it in Table 3 starting on line 283.

Some minor comments:

- On line 139-140, should 'research' be capitalised before Agenda (or both lowercase?) 

Correct, we have edited this

- On line 265, CHNRI acronym has a typo

Correct, we have edited this

- It would be useful to spell out all acronyms in their first instance (e.g., OCV, CM, etc.) for readers who are unfamiliar with the acronyms.

The acronyms have been checked and defined the first time that they are used, generally in lines 87-89.

Reviewer #2: The authors have tried to create a prioritized list of research questions to focus the limited resources and address the issues most relevant to the implementation of the Roadmap, for funding agencies and researchers to focus their efforts to fill the evidence gaps plaguing cholera-endemic countries. Intense work has been put in through various methods in achieving the research questions, however there are number of limitations and few methods lack clarity.

Most importantly the prevalence of cholera infection and the severity (mortality) due to this infection in low resource countries needs to be mentioned.

This has been updated to reflect the current situation of cholera in lines 78-81 as well as the estimated number of deaths and cases.

While seasonal episode of cholera is most seen, the intervention should focus accordingly. The population that is most affected must be described and strategies to curtail the infection needs to be explicitly explained. 

Please find relevant edits in lines 84-89.

Information on whether the vaccination has helped in reducing the infection rate and the severity must be discussed before implementing to the general population. 

The manuscript reports on an exercise to prioritize research questions to fill critical evidence gaps on interventions to control cholera. While vaccination is one of the interventions, the research questions also include other pillars. The manuscript does not make any recommendations for vaccination. Policy recommendations on the use of cholera vaccines have been made by WHO at the global level and for use at country levels by the respective national policy-making bodies. Hence, we did not feel that a discussion on the impact of one of several interventions for cholera control are relevant to this paper. 

As one of the factors mentioned is malnutrition, access to health care facilities and good nutrition, WASH, it is also important to align with the government programs and policies laid for the communicable diseases specific to the country. This is important since the countries selected are a mixed bag with different facilities and opportunities. 

This is an important point that was raised by a few interviews and discussions. We have edited to highlight the importance of open dialogue with countries to ensure full utilization and alignment to government programmes. 

Please see revised lines 345-348.

There are number of abbreviated terms that needs expansion when used for the first time in the text. Eg: GTFCC’s, OCV

These have been checked for and edited, see lines 78-81.

---

## [Decision Letter · Decision Letter 1]

21 Feb 2022

Application of the Child Health and Nutrition Research Initiative (CHNRI) methodology to prioritize research to enable the implementation of Ending Cholera: a global roadmap to 2030.

PONE-D-21-25214R1

Dear Dr. Ko,

We’re pleased to inform you that your manuscript has been judged scientifically suitable for publication and will be formally accepted for publication once it meets all outstanding technical requirements.

Kind regards,

Jai K Das

Academic Editor

PLOS ONE

Reviewers' comments:

Reviewer's Responses to Questions

**Comments to the Author**

1. If the authors have adequately addressed your comments raised in a previous round of review and you feel that this manuscript is now acceptable for publication, you may indicate that here to bypass the “Comments to the Author” section, enter your conflict of interest statement in the “Confidential to Editor” section, and submit your "Accept" recommendation.

Reviewer #1: All comments have been addressed

Reviewer #2: All comments have been addressed

2. Is the manuscript technically sound, and do the data support the conclusions?

Reviewer #1: Yes

Reviewer #2: Yes

3. Has the statistical analysis been performed appropriately and rigorously? 

Reviewer #1: Yes

Reviewer #2: Yes

4. Have the authors made all data underlying the findings in their manuscript fully available?

Reviewer #1: Yes

Reviewer #2: Yes

5. Is the manuscript presented in an intelligible fashion and written in standard English?

Reviewer #1: Yes

Reviewer #2: Yes

6. Review Comments to the Author

Reviewer #1: (No Response)

Reviewer #2: The comments raised by the reviewer has been addressed satisfactorily. There are couple of suggestion made based on the limitation identified due to methodological / logistic issues and also enhanced due to the COVID-19 pandemic. Additional information with face-to-face interviews in a subgroup on evaluating the COVID pandemic situation will be useful. It is important to revisit the work relevant to this area through a systematic literature review to help the reader in identifying research gaps. The questionnaire and other tools used should be periodically validated and standardized for accuracy.

7. PLOS authors have the option to publish the peer review history of their article (what does this mean?). If published, this will include your full peer review and any attached files.

Reviewer #1: **Yes: **Kerri Wazny

Reviewer #2: No